# The Interaction between CLSPN Gene Polymorphisms and Alcohol Consumption Contributes to Oral Cancer Progression

**DOI:** 10.3390/ijms25021098

**Published:** 2024-01-16

**Authors:** Ming-Ju Hsieh, Yu-Sheng Lo, Hsin-Yu Ho, Chia-Chieh Lin, Yi-Ching Chuang, Mu-Kuan Chen

**Affiliations:** 1Oral Cancer Research Center, Changhua Christian Hospital, Changhua 500, Taiwan; 2Doctoral Program in Tissue Engineering and Regenerative Medicine, College of Medicine, National Chung Hsing University, Taichung 402, Taiwan; 3Graduate Institute of Biomedical Sciences, China Medical University, Taichung 404, Taiwan; 4Department of Otorhinolaryngology, Head and Neck Surgery, Changhua Christian Hospital, Changhua 500, Taiwan; 5Department of Post-Baccalaureate Medicine, College of Medicine, National Chung Hsing University, Taichung 402, Taiwan

**Keywords:** single nucleotide polymorphisms, Claspin, oral cancer

## Abstract

Most disease single nucleotide polymorphisms (SNPs) are regulatory and approximately half of heritability is occupied by the top 1% of genes, with the gene-level structure varying with the number of variants associated with the most common alleles. Cancer occurrence and progression are significantly affected by Claspin (CLSPN) gene polymorphism present in the population, which alters the expression, function, and regulation of the gene. CLSPN genotypes are associated with oral cancer, but the literature on this association is limited. As a result, the goal of this study is to investigate the correlation between CLSPN genotypes and oral cancers’ development. This study will explore the presence of four CLSPN SNPs including rs12058760, rs16822339, rs535638 and rs7520495 gene polymorphisms, and analyze the expression of these genes in 304 cancer-free controls and 402 oral squamous cell carcinoma (OSCC) cases. Attempts have been made to obtain insight into the role of CLSPN gene polymorphisms in oral cancer through the analysis of this study. We demonstrated that the OSCC risk of individuals with four CLSPN SNPs relative to the wild type did not differ significantly from that of the wild type when the polymorphisms are analyzed according to individual habits. We further studied the mechanism by which CLSPN polymorphisms affect the progression of clinicopathological features in OSCC patients. The results of the degree of cell differentiation showed that compared with patients of rs7520495 SNP carrying the CC genotype, the incidence of poor cell differentiation in patients carrying the CC + GG genotype was higher (AOR: 1.998-fold; 95% CI, 1.127–3.545; p = 0.018). In particular, patients with the G genotype of rs7520495 had increased poor cell differentiation compared with patients with the C genotype (AOR: 4.736-fold; 95% CI, 1.306–17.178; p = 0.018), especially in the drinking group. On the basis of our analysis of the Cancer Genome Atlas dataset, we found that higher CLSPN levels were associated with poorer cell differentiation in oral cancers. In this study, we provide the first evidence showing that CLSPN SNPs contribute to oral cancer. Whether or not rs7520495 can be used as a confirmatory factor in the future is uncertain, but it seems likely that it can be used as an important factor in predicting recurrence, response to treatment and medication toxicity to patients with oral cancer.

## 1. Introduction

According to the data provided by the World Health Organization, head and neck cancer is a significant global health issue. Approximately 660,000 people are diagnosed with head and neck cancer each year, and about 325,000 individuals die from this disease annually [1]. Head and neck cancer is the sixth most common cancer in the world and is more common in men. Most countries have a high mortality rate of nearly 50% for oral squamous cell carcinoma (OSCC), an aggressive tumor with a poor prognosis. At the time of their initial detection, approximately half of all oral cancers diagnosed are advanced. [2,3,4]. OSCC is a cancer of the mucosal epithelial cells of the upper aerodigestive tract and is the most common type accounting for more than 90% of head and neck cancer cases. The oral cavity can be affected by OSCC in a variety of locations, including the lips, tongue, gums, buccal mucosa, tongue floor, hard palate, and alveolar ridge [5]. For the development of OSCC, there are many risk factors that may contribute to the development of this disease, such as smoking, alcohol consumption, chewing betel nut, human papillomavirus (HPV) infection, Epstein–Barr virus infections, and exposure to other environmental factors [6,7,8,9,10]. Alcohol in particular also promotes the progression and aggressiveness of existing cancers. Long-term exposure to ethanol in OSCC increases the number of cancer stem cells, and heavy drinkers induce Toll-like receptors that promote chronic inflammation, tumor cell migration, and invasion [11,12]. There is evidence that ethanol induces apoptosis and has the ability to arrest cell division in human neuroblastoma cells [13]. Moreover, OSCC occurrence is affected by a synergistic interaction between genetic risk factors and environmental factors that may contribute to the occurrence of OSCC [14]. According to genome-wide or targeted gene association studies, studies have shown that single nucleotide polymorphisms have a significant association with cancer development [15,16,17].

Claspin (CLSPN) is a Chk1 (checkpoint kinase 1)-interacting protein found in Xenopus oocyte extracts and plays a critical role in the phosphorylation and activation of Chk1 by the downstream kinase ATR in response to inhibition of DNA synthesis [18]. As well as DNA polymerases, CLSPN also interacts with a number of replication factors, such as ATR, Chk1, Cdc7 kinases, and Cdc45, suggesting that CLSPN participates in the replication process at replication forks and may even play a role in the replication initiation process itself [19]. Previous studies have demonstrated that mutations in CLSPN phosphorylation sites inhibit CLSPN-Chk1 interactions in vivo, impair Chk1 activation, and induce premature chromatin condensation, thereby confirming defective replication checkpoints [20]. In addition to its function in repairing DNA damage, CLSPN plays a role in the regulation of the cell cycle and DNA repair. It plays a crucial role in the S phase and G2/M phase of the cell cycle, regulating DNA replication and cell division [18,21,22]. Furthermore, elevated expression of CLSPN is closely related to the induction of immune cell infiltration, tumor mutation burden, microsatellite instability, mismatch repair, and DNA methylation in various cancer types [23]. In recent years, researchers have begun to study the role of the CLSPN gene in cancer development, specifically after finding high levels of CLSPN in cancers as diverse as breast, ovarian, cervical, glioma, non-small cell lung cancer and renal cell carcinoma [24,25,26,27,28]. 

Earlier studies have shown that ATR-CHK1 upregulation leads to the accumulation of OSCC cells in the G2 phase as a response to ionizing radiation, which makes the cells resistant to radiation [29]. There is a crucial role that CLSPN plays in the transmission of signals between ATR and Chk1, that are directed to repair the cell cycle and induce apoptosis [30]. The molecular mechanism of cordycepin’s action may be related to its ability to prolong the G2/M phase arrest and double-stranded DNA breaks in oral cancer cells, and it may extend the G2/M phase arrest through its modulation of ATR-Chk1 [31]. Therefore, Chk1 inhibition may be a promising strategy for treating these diseases [32]. Chk1 inhibitors, such as UCN-01, have been successfully used in clinical trials to treat certain types of cancer. In particular, UCN-01 exhibited potent in vivo anti-tumor activity when administered to mice carrying p53-mutant HNSCC xenografts [33]. Additionally, Chk1 inhibitors have shown promise in mitigating the side effects of chemotherapy and radiation therapy [34,35,36]. Research evidence suggests that abrogating the G2/S checkpoint with Chk1 inhibitors is likely to expose head and neck cancers to ionizing radiation as well as other DNA-damaging agents such as cisplatin when the G2/S checkpoint is ablated [37,38]. Moreover, the downregulation of Claspin was found in head and neck squamous cell carcinoma (HNSCC) after triple therapy [39]. 

CLSPN gene polymorphism refers to the DNA sequence variations or mutations of the CLSPN gene that exist within a population. These gene polymorphisms may impact the expression, function, or regulation of the CLSPN gene, thereby exerting significant effects on cancer occurrence and progression [25]. However, the mechanisms leading to OSCC occurrence and single nucleotide polymorphisms (SNPs) remain unclear. In the present study, we aimed to investigate whether the SNP of CLSPN correlates with the development and clinicopathological characteristics of OSCC in Taiwanese individuals.

## 2. Results

### 2.1. Basic Characters between the Non-OSCC and OSCC Groups

This study collected and analyzed data from two groups of OSCC patients and cancer-free controls, as shown in Table 1. The age distribution was similar between the two groups (*p* = 0.421), and as for the distribution of betel nut chewing, smoking, and alcohol consumption, the percentages showed significant differences between controls and OSCC patients (*p* < 0.001). Tumor characteristics of the OSCC cohort, including tumor stage, TNM status, and histopathological grade, are also published in Table 1. There was a predistribution of non-metastasis to lymph nodes (71.4%), and there was no distant organ metastasis detected (95.8%). There is also evidence in Table 1 that indicates that approximately 83.3% of OSCC cases are characterized by a moderate to poor degree of cellular differentiation.

### 2.2. Distribution Frequency of CLSPN SNPs between Non-OSCC and OSCC Groups

To explore the genotype distribution and association between OSCC and CLSPN SNPs, four SNPs of the CLSPN gene (rs12058760, rs16822339, rs535638, and rs7520495) were evaluated in the control group and OSCC group, respectively, as shown in Table 2. The OSCC risk of individuals with CLSPN gene polymorphisms rs12058760, rs16822339, rs535638 and rs7520495 relative to the wild type did not differ significantly from that of the wild type when the polymorphisms are analyzed according to individual habits. Meanwhile, we also analyzed the genetic parameters of drinkers/non-drinkers, betel quid chewers/non-betel quid chewers and smokers/non-smokers in the control group and OSCC group. The results are presented in Appendix A, respectively, for drinkers/non-drinkers (Appendix A), betel nut chewers/non-betel nut chewers (Appendix A), and smokers/non-smokers (Appendix A). According to the results of our analysis, it was known that CLSPN SNPs are not associated with susceptibility to oral cancer among various groups of environmental factors.

### 2.3. CLSPN Polymorphism Can Affect the Mechanism of Progression Clinicopathological Characteristics in Patients with OSCC

We then investigated the correlation between genotype and clinical and pathological characteristics in OSCC patients, looking for links between the CLSPN gene polymorphisms rs12058760, rs16822339, rs535638 and rs7520495 genotypes. Following adjustment for other variables, adjusted odds ratios (AORs) and their 95% confidence intervals (CIs) were estimated using multiple logistic regression models. The results in Table 3 indicate that the different distributions of CLSPN SNP rs7520495 (including CC and CG + GG) are not related to the clinical stage, tumor size, lymph node invasion and distant metastasis status. In particular, the results of the degree of cell differentiation showed that compared with patients carrying the CC allele, the incidence of poor cell differentiation in patients carrying the CG + GG allele was higher (AOR: 1.998-fold; 95% CI, 1.127–3.545; p = 0.018). On the other hand, according to the results of the analysis, we observed no association between CLSPN SNPs (rs12058760, rs16822339 and rs535638) and recruited OSCC susceptibility.

### 2.4. Combined and Interactive Effects of CLSPN rs7520495 and Habitual Risk Factors on OSCC Progression 

As further analysis of rs7520495 was conducted, we were able to determine the relationship between different genotypes in patients and various risk factors associated with the OSCC, such as alcohol consumption, chewing of betel quid, and smoking. The results in Table 4 show that in the drinking group, compared with patients carrying the CC allele, the incidence of poor cell differentiation in patients carrying the CG + GG allele was higher (AOR: 4.736-fold; 95% CI, 1.306–17.178; p = 0.018). In contrast, we found no association between CLSPN rs7520495 and OSCC susceptibility in non-drinking groups. As a result of our findings, we found an association between the presence of at least one minor allele of CLSPN rs7520495 SNP between alcohol consumption.

### 2.5. Clinical and Functional Insights from CLSPN to OSCC

The characteristics of the clinical outcome of the CLSPN gene and the OSCC were analyzed from the TCGA database. We found that CLSPN genes are related to clinicopathological characteristics based on the results in Figure 1. Neither clinical stage, tumor size, nor lymph node status differed (*p* = 0.7647; *p* = 0.3324; *p* = 0.8441). Meanwhile, in terms of the degree of cell differentiation, there was a significant difference between well and moderate states (*p* = 0.0076) as well as between well and poor states (*p* = 0.0035) in terms of cell differentiation. This illustrates the importance of demonstrating a genetic association between this gene and the OSCC.

## 3. Discussion

According to our findings in this study, we found that the CLSPN rs7520495 polymorphism is significantly associated with a clinical cell differentiation status, particularly for patients with the G allele. As a final observation, through the analysis of individual habits, we observed that the CLSPN gene polymorphism rs7520495 exacerbated the deterioration in cell differentiation when combined with alcohol consumption and clinical cell differentiation status.

As most disease SNPs are regulatory, disease-associated SNPs do not directly affect target genes. The gene-level structure varied with the number of variants associated with the most common alleles, and approximately half of the SNP heritability associated with all genes was accounted for by the top 1% of genes [40]. Structure-based effects of diseases-associated SNPs may depend on gene-level factors. Additionally, these results provide further evidence for the role of regulatory SNPs in disease pathogenesis. Furthermore, SNPs in these regions can alter gene expression or alter gene–molecule interactions leading to complex diseases [16]. A CLSPN mutation c.1574A>G has been demonstrated to reduce CLSPN expression and activate Chk1, which may affect the CLSPN structure and function in breast cancer [25]. There is evidence that CLSPN, which has been found to express the I783S missense mutation that inhibits its ability to mediate Chk1 phosphorylation after DNA damage, may be associated with tumorigenesis [24]. As a result of further study of the Chk1 binding domain of phosphorylated CLSPN, it has been established that Cdc7 is necessary for the interaction between CLSPN and Chk1 in human cancer cells [19]. Based on the above, it can be demonstrated that the mutation of the CLSPN gene results in the loss of its regulatory function, which in turn affects the phosphorylation of chk1 and induces cancer development. 

In this study, CLSPN gene polymorphisms were analyzed based on individual habits to explore the relationship between the wild type and OSCC. After adjusting other variables, a multiple logistic regression model was used to estimate the AOR and its 95% CI. The risk of OSCC in individuals with CLSPN gene polymorphisms is not significantly different from that in the wild type. However, we further explored the correlation between clinical manifestations and genetic variants of CLSPN SNP. Among the clinical characteristics items, especially the results of cell differentiation status, the highly differentiated status of cells in patients with G genotype increased 1.998-fold compared with patients with SNP C genotype rs7520495. Previous findings demonstrate that high CLSPN expression levels in vivo are significantly associated with HR-HPV infection and lesion grade in histological and cytological samples [26]. As a consequence of the above findings, it can be concluded that CLSPN shows high sensitivity for cervical intraepithelial neoplasia (CIN) and squamous cell carcinoma (SCC). Related to our results, patients with the CLSPN G genotype had a higher degree of cellular differentiation than patients with the SNP C genotype rs7520495. Notably, the statistically significant associations were found between alcohol consumption and oral cancer, even when compared to the effects different alleles had on the habits of the different individuals [41,42,43,44,45]. As in previous studies, DNA modifications related to acetaldehyde have also been found in the oral cells of humans who have consumed alcohol in the past, suggesting that the metabolism of alcohol in the mouth is an independent risk factor for developing cancer [46]. The highly differentiated state of cells in G genotype patients is 4.736-fold higher than that in patients with SNP C genotype rs7520495, exclusively among patients who drink alcohol. Our demonstration that CLSPN polymorphisms have strong effects and significant differences in susceptibility to alcohol consumption in oral cancer patients, consistent with the above studies. 

Researchers identified risk-associated SNPs by conducting genome-wide association studies, which can be used to screen subjects in epidemiological studies to establish relationships between SNPs and tumors [47]. In order to identify molecular aberrations at the DNA, RNA, protein, and epigenetic levels in a large collection of human tumor tissues, TCGA research network profiles and analyzes a large number of tumor samples [48]. Human cancer incidence and/or prevalence are influenced by the interaction between specific allelic variants and environmental factors resulting in highly modulated disease susceptibility due to SNP genetic variation [49]. In addition, the cytopathological moderate/highly differentiated status of oral cancer was observed according to the TCGA database. This study also found that there is such a relationship between CLSPN SNP rs7520495 and the above clinical pathology characteristics, especially in the drinking group.

However, our study possesses some limitations worth discussing. There is uncertainty about whether these results can be observed in any other ethnic group than Taiwanese given that all subjects in this study were of regional descent (Taiwanese). Further studies are needed to confirm our findings on CLSPN SNPs in different ethnic OSCC cohorts. On the other hand, the group sample size is insufficient, which may affect results statistically. As the study population selection, a person with a habit of cigarette smoking, alcohol consumption or betel chewing makes up less than 10% of the control group; this number is too small compared with the OSCC patient group. For this reason, we analyzed CLSPN variants in control and OSCC cases by adjusting for personal habits. After adjusting for personal habits, there was no significant difference in the AOR between the control group and the OSCC patient group, suggesting that CLSPN variants rs7520495 may be related to the occurrence of personal habits.

## 4. Materials and Methods

### 4.1. Patients and Specimens

The samples for this study were collected at Changhua Christian Hospital. Regulatory approval for this study was obtained from the Institutional Review Board (IRB) at Changhua Christian Hospital (CCH) under the number 230603. The research group included 402 patients diagnosed with OSCC and 304 cancer-free patients in the control group in Changhua Christian Hospital from 2014 to 2023. A total of 706 cases were collected for this study, and all patients participating in the study signed a written informed consent form before the start of the project. There was no geographical difference between the study populations and they all lived in Han Chinese communities. We obtained statistical data on age and personal habits (including betel nut chewing, smoking and alcohol consumption) from medical documents. Additionally, AJCC No. 8 was also used to discuss the judgment of the clinical stage, tumor/lymph node/metastasis (TNM) stage and degree of cell differentiation [50]. For CLSPN polymorphisms, venous blood samples were collected by the investigator and stored in tubes containing K3-Ethylene diamine tetra acetic acid (EDTA). The blood samples were then cryogenically centrifuged and stored in a −80 °C laboratory freezer for analysis.

### 4.2. Functional CLSPN SNP Selection

Due to the few references to CLSPN SNPs, we selected other CLSPN SNPs in Asian populations by using the ABI probe database and filtered them using linkage disequilibrium (LD) and minor allele frequency (MAF). The CLSPN SNPs were selected using an ABI SNPbrowser, and we then excluded LD sites through the LDlink website. Subsequently, we excluded the MAF lower of the genetic loci by the National Institutes of Health Variation Viewer. This eliminates options with LD-LINK scores higher than 0.8 and MAF percentages lower than 10%. After eliminating the above conditions, three CLSPN polymorphism sites were selected, namely rs12058760, rs16822339 and rs7520495. Previous studies have shown that CLSPN rs535638 is significantly associated with familial breast cancer and glioma [25], and we included CLSPN rs535638 in the candidate list. Finally, four polymorphisms were obtained: rs12058760, rs16822339, rs535638 and rs7520495, and the MAF of each polymorphism were 12.6%, 30.3%, 48.3% and 39.5%, respectively. The four CLSPN SNPs rs12058760 (C/G), rs16822339 (A/C), rs535638 (C/T), and rs7520495 (C/G) obtained from the above analysis were included in the analysis model.

### 4.3. DNA Extraction and Analyzed CLSPN SNP with Real-Time PCR

Similar to our previous research, we used DNA extraction, preservation, and analysis techniques [43,51]. Whole blood samples were collected from patients into sterile tubes containing EDTA, which were immediately centrifuged and stored at −80 °C. The genomic DNA was extracted from peripheral blood leukocytes using a QIAamp DNA blood mini kit, and then dissolved in TE buffer and stored at −20 °C. Quantification was based on measuring the optical density at a wavelength of 260 nm. The four polymorphisms rs12058760 (C/G), rs16822339 (A/C), rs535638 (C/T), and rs7520495 (C/G) of the CLSPN gene potential were determined by real-time quantitative PCR using the ABI StepOne real-time PCR system (Applied Biosystems, Foster City, CA, USA) and were analyzed using StepOne Software v2.3. The sum of 2.5 µL TaqMan Genotyping Master Mix, 0.125 µL TaqMan probe mix, and 30 ng genomic DNA was used to create each reaction, resulting in a final volume of 5 µL. Real-time PCR was conducted with an initial denaturation step at 95 °C for 10 min, followed by 40 amplification cycles at 95 °C for 15 s and 60 °C for 1 min.

### 4.4. Bioinformatics Analysis of CLSPN Expression

This study aimed to explore potential associations between CLSPN expression and OSCC clinical status using the University of California, Santa Cruz Xena functional genomics browser [52]. We analyzed this question using HNSCC data obtained from the TCGA database [48]. This study classifies head and neck squamous cell carcinoma into different subgroups according to tumor stage and TNM status, and compares CLSPN mRNA levels between the subgroups.

### 4.5. Statistical Analysis

We used IBM SPSS Statistics v22.0 (IBM, Armonk, NY, USA) to conduct the analyses in our study, similarly to previous papers [51]. First, the demographic and laboratory data between the non-OSCC group and the OSCC group were shown using descriptive analysis including mean, standard deviation (SD) and percentage, and evaluated using a Mann–Whitney U test to test the difference between the two groups. We were able to estimate the adjusted odds ratios (AORs) using a multiple logistic regression model in SPSS statistical analysis after controlling betel nut chewing, alcohol and tobacco consumption to exclude possible influencing environmental factors. Logistic regression models were then used to analyze the AORs and associated 95% CI of the CLSPN SNP polymorphism distribution between non-OSCC and OSCC populations. We further divided OSCC patients into non-drinkers and drinkers, and analyzed the correlation between CLSPN SNP rs7520495 and OSCC clinicopathological characteristics, in order to generate an AOR with 95% confidence intervals. CLSPN level variation in TCGA’s HNSCC data set was compared with the Mann–Whitney U test. A *p*-value below 0.05 was defined as of statistical significance.

## 5. Conclusions

In summary, according to our experimental results, we found that compared with patients carrying the CC genotype of rs7520495 SNP, patients carrying the CC+GG genotype correlated to higher rates of poor cell differentiation. We then separately adjusted different personal habits for subsequent analysis and found that patients with the G genotype of rs7520495 had poor cellular differentiation compared with patients with the C genotype among alcohol drinkers. Our study is the first to provide evidence on the interactive effect of CLSPN gene polymorphisms and alcohol drinking on the progression of oral cancer. Whether or not rs7520495 can be used as a confirmatory factor in the future is uncertain, but it seems likely that it can be used as an important factor in predicting recurrence, response to treatment and medication toxicity to patients with oral cancer.

## Figures and Tables

**Figure 1 ijms-25-01098-f001:**
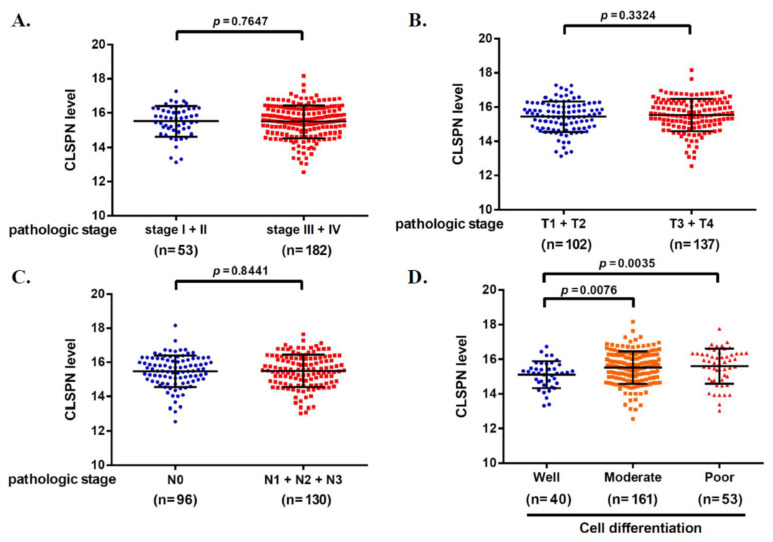
The CLSPN expression in the OSCC with different grades according to the TCGA database. The correlation between CLSPN expression level and (**A**) clinical stage, (**B**) tumor size, (**C**) lymph node metastasis, and (**D**) cell differentiation in OSCC. Results with a *p*-value of less than 0.05 were considered statistically significant.

**Table 1 ijms-25-01098-t001:** The distributions of demographical characteristics and clinical parameters in 304 controls and 402 cases with OSCC.

Variable	Control (N = 304)	Patients (N = 402)	*p* Value
**Age (yrs.)**	53.94 ± 7.74	54.07 ± 9.57	
>54	167 (54.9%)	197 (49.0%)	0.421
≤54	137 (45.1%)	205 (51.0%)	
**Betel nut chewing**			
No	293 (96.4%)	144 (35.8%)	<0.001 *
Yes	11 (3.6%)	258 (64.2%)	
**Cigarette smoking**			
No	281 (92.4%)	90 (22.4%)	<0.001 *
Yes	23 (7.6%)	312 (77.6%)	
**Alcohol drinking**			
No	296 (97.4%)	286 (71.1%)	<0.001 *
Yes	8 (2.6%)	116 (28.9%)	
**Stage**			
I + II		208 (51.7%)	
III + IV		194 (48.3%)	
**Tumor T status**			
T1 + T2		262 (65.2%)	
T3 + T4		140 (34.8%)	
**Lymph node status**			
N0		287 (71.4%)	
N1 + N2 + N3		115 (28.6%)	
**Metastasis**			
M0		385 (95.8%)	
M1		17 (4.2%)	
**Cell differentiation**			
Well differentiated		67 (16.7%)	
Moderately or poorly differentiated		335 (83.3%)	

N: number. * *p* value < 0.05 as statistically significant.

**Table 2 ijms-25-01098-t002:** The distribution of genotype frequencies in CLSPN SNPs in cases of OSCC group.

Variable	Control (N = 304)	Patients (N = 402)	AOR ^a^ (95% CI)	*p* Value
**rs12058760**				
CC	268 (88.2%)	360 (89.5%)	1.000 (reference)	
CG	35 (11.5%)	40 (10.0%)	0.800 (0.389–1.649)	0.546
GG	1 (0.3%)	2 (0.5%)	4.120 (0.299–56.84)	0.290
CG+GG	36 (11.8%)	41 (10.5 %)	0.888 (0.444–1.778)	0.737
**rs16822339**				
AA	182 (59.9%)	216 (53.7%)	1.000 (reference)	
AC	103 (33.9%)	163 (40.5%)	1.128 (0.714–1.782)	0.607
CC	19 (6.3%)	23 (5.8%)	1.444 (0.601–3.468)	0.411
AC + CC	122 (40.1%)	186 (46.3%)	1.171 (0.758–1.809)	0.476
**rs535638**				
CC	209 (68.8%)	247 (61.4%)	1.000 (reference)	
CT	85 (28.0%)	145 (36.1%)	1.329 (0.835–2.117)	0.230
TT	10 (3.3%)	10 (2.5%)	1.027 (0.279–3.783)	0.968
CT + TT	95 (31.3%)	155 (38.6%)	1.301 (0.828–2.044)	0.255
**rs7520495**				
CC	121 (39.8%)	114 (28.4%)	1.000 (reference)	
CG	143 (47.0%)	227 (56.5%)	1.326 (0.823–2.134)	0.246
GG	40 (13.2%)	61 (15.1%)	1.542 (0.785–3.031)	0.209
CG + GG	183 (60.2%)	288 (71.6%)	1.370 (0.870–2.158)	0.174

N: number. ^a^ The adjusted odds ratio (AOR) with their 95% confidence intervals were estimated by multiple logistic regression models after controlling for betel nut chewing, alcohol and tobacco consumption.

**Table 3 ijms-25-01098-t003:** Clinical statuses and CLSPN rs7520495 genotype frequencies in cases of OSCC group.

Variable	CLSPN (rs7520495)			
	CC (N = 114)	CG + GG (N = 288)	AOR ^a^ (95% CI)	*p* Value
**Clinical stage**				
Stage I/II	60 (52.6%)	148 (51.4%)	1.000 (reference)	0.843
Stage III/IV	54 (47.4%)	140 (48.6%)	1.047 (0.667–1.643)	
**Tumor size**				
T1 + T2	75 (65.8%)	187 (64.9%)	1.000 (reference)	0.982
T3 + T4	39 (34.2%)	101 (35.1%)	1.005 (0.626–1.615)	
**Lymph node metastasis**				
No	85 (74.6%)	202 (70.1%)	1.000 (reference)	0.442
Yes	29 (25.4%)	86 (29.9%)	1.221 (0.734–2.031)	
**Distant metastasis**				
No	110 (96.5%)	275 (95.5%)	1.000 (reference)	0.987
Yes	4 (3.5%)	13 (4.5%)	1.010 (0.305–3.346)	
**Cell differentiation**				
Well differentiated	27 (23.7%)	40 (13.9%)	1.000 (reference)	0.018 *
Moderately or poorly differentiated	87 (76.3%)	248 (86.1%)	1.998 (1.127–3.545)	

N: number. ^a^ The adjusted odds ratio (AOR) with their 95% confidence intervals were estimated by multiple logistic regression models after controlling for betel nut chewing, alcohol and tobacco consumption. * *p* value < 0.05 as statistically significant.

**Table 4 ijms-25-01098-t004:** The association between CLSPN rs7520495 genotype frequency and clinical status with and without alcohol drinker.

Variable	CLSPN (rs7520495)							
	Alcohol Drinkers (N = 116)				Non-Alcohol Drinkers (N = 286)			
	CC (N = 12)	CG + GG (N = 104)	AOR ^a^ (95% CI)	*p* Value	CC (N =102)	CG + GG (N =184)	AOR ^a^ (95% CI)	*p* Value
**Clinical stage**								
Stage I/II	7 (58.3%)	53 (51.0%)	1.000 (reference)	0.585	53 (52.0%)	95 (51.6%)	1.000 (reference)	0.970
Stage III/IV	5 (41.7%)	51 (49.0%)	1.431 (0.413–4.783)		49 (48.0%)	89 (48.4%)	1.009 (0.620–1.643)	
**Tumor size**								
T1 + T2	9 (75.0%)	64 (61.5%)	1.000 (reference)	0.371	66 (64.7%)	123 (66.8%)	1.000 (reference)	0.731
T3 + T4	3 (25.0%)	40 (38.5%)	1.865 (0.476–7.311)		36 (35.7%)	61 (33.2%)	0.914 (0.548–1.525)	
**Lymph node metastasis**								
No	8 (66.7%)	73 (70.2%)	1.000 (reference)	0.873	77 (75.5%)	129 (70.1%)	1.000 (reference)	0.333
Yes	4 (33.3%)	31 (29.8%)	0.899 (0.244–3.312)		25 (24.5%)	55 (29.9%)	1.314 (0.756–2.283)	
**Distant metastasis**								
No	11 (91.7%)	97 (93.3%)	1.000 (reference)	0.852	99 (97.1%)	178 (96.7%)	1.000 (reference)	0.891
Yes	1 (8.3%)	7 (6.7%)	0.811 (0.090–7.290)		3 (2.9%)	6 (3.3%)	1.104 (0.269–4.521)	
**Cell differentiation**								
Well differentiated	5 (41.7%)	14 (13.5%)	1.000 (reference)	0.018 *	22 (21.6%)	26 (14.1%)	1.000 (reference)	0.120
Moderately or poorly differentiated	7 (58.3%)	90 (86.5%)	4.736 (1.306–17.178)		80 (78.4%)	158 (85.9%)	1.649 (0.878–3.095)	

N: number. ^a^ The adjusted odds ratio (AOR) with their 95% confidence intervals were estimated by multiple logistic regression models after controlling for betel nut chewing and tobacco consumption.* *p* value < 0.05 as statistically significant.

## Data Availability

The datasets generated for this study are available on request to the corresponding authors.

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
