# Peer review of "The Interaction between CLSPN Gene Polymorphisms and Alcohol Consumption Contributes to Oral Cancer Progression"

_ijms, 2024, doi:10.3390/ijms25021098_

Round 1
Reviewer 1 Report
Comments and Suggestions for Authors
In the present manuscript, it indicated that rs535638 and rs7520495 SNPs in Claspin (CLSPN) gene has a risk of oral squamous cell carcinoma (OSCC) by compare the frequency of 4 polymorphisms (rs12058760, rs16822339, rs535638 and rs7520495 SNPs) between controls and OSCC patients. However, the manuscript is not well-written. I recommend that this paper accepted after minor revision.
1. It is not well-written about 4 polymorphisms (rs12058760, rs16822339, rs535638 and rs7520495 SNPs. Does these SNPs affect the protein or expression of CLSPN? Also, it is better to mention the minor allele frequency (MAF) of 4 SNPs.
2. It is unclear to focus these 4 SNPs. It should mention about the reason.
3. In line 113-114, it mentioned that after adjusting for the risks associated with smoking, eating betel nuts and drinking alcohol factors, we were able to estimate. However, it is not well-written about the adjusting method.
4. Many CLSPN are written incorrectly in CLSNP.
5. An incorrect line on Betel nut chewing in Table 1 has been added.
Reviewer 2 Report
Comments and Suggestions for Authors
Dear Authors,
Although there is a lot of work done behind the manuscript entitled "Variations in CLSPN polymorphisms are associated with increased risk of oral cancer in drinkers", it looks like a draft and needs major revision before resubmission.
Here are my recommendations:
Line 49: The following statement is incorrect and should be revisited with care: "the most common type of cancer in men". Also"approximately 10% of patients dying within six months of diagnosis and high mortality rates in advanced 50 stages" refers to one nationwide study, not a worldwide evaluation.
Line 68: Reference 18 was not published in.." published in Xenopus oocyte extracts"
Lines 93: There is a mistake in the manuscript workflow After the Introduction, there is Results Chapter. Materials and methods chapter starts with line 294.
Line 105: In Table 1 can you detail why the study population was divided by age 54?
Line 182: The Legend of Figure 1 contains the Aim of the study.
Line 299 Can you detail the selection of the control-group ? Were all these patients cancer-free regarding the oral cavity or the whole body?
Lines 311-313 As this is a part of the Material and Methods chapter it is expected to describe the method, not to state some results and an Internet link with no connection.
Lines 315-316: The description of the methods needs to be detailed in this manuscript . "Similar to our previous research, we used extraction, preservation, and analysis techniques"
Reviewer 3 Report
Comments and Suggestions for Authors
overall a good study. However, the presentation specially the tables must be improved
Round 2
Reviewer 2 Report
Comments and Suggestions for Authors
Dear Authors,
The manuscript has been corrected and I recommend its acceptance.
Best regards!
Author Response
Thank you for your comment